# Changes in Cognition and Decision Making Capacity Following Brain Tumour Resection: Illustrated with Two Cases

**DOI:** 10.3390/brainsci7100122

**Published:** 2017-09-24

**Authors:** Katie Veretennikoff, David Walker, Vivien Biggs, Gail Robinson

**Affiliations:** 1Neuropsychology Research Unit, The University of Queensland, Brisbane 4072, Australia; katie.veretennikoff@uqconnect.edu.au; 2BrizBrain and Spine, The Wesley Hospital, Brisbane 4066, Australia; dwalker@brizbrain.com.au (D.W.); vivienb@brizbrain.com.au (V.B.)

**Keywords:** brain tumours, cognition, capacity, decision making, neuropsychology

## Abstract

Changes in cognition, behaviour and emotion frequently occur in patients with primary and secondary brain tumours. This impacts the ability to make considered decisions, especially following surgical resection, which is often overlooked in the management of patients. Moreover, the impact of cognitive deficits on decision making ability affects activities of daily living and functional independence. The assessment process to ascertain decision making capacity remains a matter of debate. One avenue for evaluating a patient’s ability to make informed decisions in the context of brain tumour resection is neuropsychological assessment. This involves the assessment of a wide range of cognitive abilities on standard measurement tools, providing a robust approach to ascertaining capacity. Evidence has shown that a comprehensive and tailored neuropsychological assessment has greater sensitivity than brief cognitive screening tools to detect subtle and/or specific cognitive deficits in brain tumours. It is the precise nature and severity of any cognitive deficits that determines any implications for decision making capacity. This paper focuses on cognitive deficits and decision making capacity following surgical resection of both benign and malignant, and primary and secondary brain tumours in adult patients, and the implications for patients’ ability to consent to future medical treatment and make decisions related to everyday activities.

## 1. Introduction

Changes in cognition often occur as a consequence of brain tumours and their treatment, including surgical resection, which has implications for decision making capacity. The ability to make informed decisions is often overlooked in patients with brain tumours, despite cognitive alterations being common [1]. The core features of making an informed decision about health, lifestyle, finances, or legal matters requires that an individual, including those with a brain tumour, have the ability to understand the relevant information, apply abstract reasoning, make a decision, and then effectively communicate that decision [2]. In addition, patients with brain tumours experience a range of physical and psychiatric changes that can cause significant disability that impacts this process [3]. Capacity refers to an individual’s ability to make a specific decision at a particular point in time, and does not refer to the ability to make all decisions [4,5]. Thus, a patient may have the capacity to consent to medical treatment, but not have capacity to manage their finances, for example. Decision making is underpinned by a range of cognitive skills and neuropsychological assessment is important to assess these underlying skills, as well as provide insight into potential barriers for the patient to provide informed consent, such as impaired language [2]. Understanding a patient’s cognitive strengths and weaknesses can help identify targets for cognitive rehabilitation, which may increase decision making capacity. Decision making capacity will often need to be an ongoing process in cases where the brain tumour is aggressive and progressive, as it can lead to progressive and sometimes rapid decline in cognitive functioning [1]. From an ethical perspective, patients lacking capacity need to be protected, and an evidence-based approach to determine capacity is essential [6]. Neuropsychological assessment as part of capacity assessment provides an objective process that is less reliant on subjective clinical opinion of a patient’s capacity to give consent at a specified point in time for a specific decision [7]. Whilst capacity is also an important consideration in the paediatric brain tumour population, capacity assessments are hampered by legislative regulations establishing an age where individuals are presumed to have capacity to provide consent [8]. Although a young person’s cognitive development can influence their ability to understand their illness and treatment options [9], decision making in the paediatric population often falls to the parents or guardians of the patient [10]. As such, the current review will focus on the role of cognition in decision making in adults following brain tumour resection of all tumour types, including benign and malignant, and primary and secondary (i.e., metastases). 

## 2. Decision Making in Brain Tumour Patients

Adults are presumed to have the capacity to made decisions, except when it can be demonstrated that they lack capacity [2]. However, a patient should not be assumed to lack capacity based solely on disability, their medical condition, an apparent inability to communicate, or when they make a decision that the treating medical practitioner disagrees with [11]. Although it is impractical and unnecessary to conduct formal capacity assessments for every patient, these assessments should be considered when a patient is displaying risk factors for impaired decision making ability [12]. Such risk factors include a diagnosis of an organic mental disorder (e.g., brain tumour) and/or when the patient’s decision is contradictory to what most people would choose, and contradictory to a previously expressed attitude by that individual [12]. However, capacity assessments are unlikely to be triggered by making a decision that aligns with advice, despite evidence that a significant proportion of patients have reduced capacity to make decisions about their treatment [13]. The inclusion of brief and tailored neuropsychological assessment both before and after surgery can help to identify changes in cognition that may limit decision making capacity, thereby triggering a formal capacity assessment. Those involved in the care of those with brain tumours should remain vigilant to indicators that the patient’s decision making ability is compromised. Family members have the ability to play a crucial role in noticing these risks, which could help to ensure capacity assessments are initiated appropriately. 

### 2.1. Tumour and Treatment Factors Affecting Capacity 

Although impairments in medical decision making can occur in any brain tumour patient [1,14,15], there are some variables that particularly impact decision making ability. Tumour grade is negatively associated with capacity to consent to medical intervention, with more than one third of patients with World Health Organisation (WHO) grade IV tumours reported to have decision making incapacity relating to neurosurgery [15]. Patients with reduced decision making capacity tend to have higher levels of cognitive impairment, which is linked with tumour factors including tumour size, oedema and mass effect, with impairments not always being alleviated following surgical resection [16]. Prior to surgery, meningioma patients tend to experience changes in memory, attention and executive functioning [17]. Following surgery, cognitive functioning in these domains tends to improve, although still remain impaired compared healthy adults [17]. Processing speed and reaction time can also be differentially affected in these patients, both pre- and post-surgery [18]. Frontal meningioma patients tend to have particular impairments in working memory, executive functioning, and attention, and impairment in the latter two domains often persists post-surgery [19]. 

In glioma patients, left hemisphere tumours are associated with poorer verbal performance, including verbal learning and verbal intelligence, language [20], and verbal memory [21]. Patients with right hemisphere tumours experience greater difficulty with visual-perceptual skills, such as facial recognition and visuo-constructional skills [20]. Executive functioning, verbal memory and visuo-spatial memory can also be impaired in glioma patients, with tumour size and higher grade being associated with greater impairments in memory and executive functioning [16]. Prior to surgery, low-grade glioma patients show less cognitive changes compared to high-grade gliomas [22,23,24]. Following surgery, low-grade glioma patients frequently experience declines in cognition, although cognition tends to show recovery within a few months of surgery [16,25]. For patients with brain metastases, even at first presentation, a high rate of cognitive dysfunction is common [26], as is decision making capacity compromise [14]. Cognitive changes in patients with brain metastases are more associated with the size of the metastases, rather than the number of metastases [27]. 

Treatment for brain tumours can impact on capacity to make decisions. Effects and side-effects of medications and other adjuvant medical treatments for brain tumours, such as chemotherapy and radiotherapy can impact on cognitive functioning, which may impair ability to provide informed consent [28,29]. Specifically, treatments can result in changes in concentration, memory, executive functioning, and language skills, such as word finding that can impact communication [29]. Chemotherapy is especially associated with impairments in memory and concentration issues and, although these tend to be subtle [30], there may still be an impact on decision making capacity. Radiotherapy in particular has been a focus of research, with evidence suggesting whole brain radiation treatment (WBRT) is associated with high rates of neurocognitive decline; specifically, almost 90% of patients experience decline at one year [31] and memory changes are common [32]. Despite WBRT causing neurocognitive changes, the neurocognitive effects of the progression of brain metastases themselves results in greater cognitive dysfunction [33]. Comparatively, primary brain tumours are treated with focused radiation (i.e., external beam radiation therapy), which reduces the risk of more global cognitive deficits [34]. Consequently, treatment effectiveness is of particular relevance to medical treatment choice in patients with brain tumours, especially those with infiltrative or metastatic tumours. The treatment for brain cancer is rarely curative and can cause cognitive alterations, which means that patients need to balance the possibility of extending life with treatment burden and side-effects, including cognitive dysfunction [6]. 

### 2.2. Medical Decision Making

Capacity to make medical decisions involves several components: (i) understanding relevant information regarding the decision about treatment; (ii) appreciating the significance of the information provided about the illness and its treatment, particularly in relation to one’s own situation; (iii) reasoning in an abstract manner about the information provided in a way that allows for the individual to compare the risks and benefits of treatment options; and (iv) expressing a choice relating to treatment [35]. This process is seldom undertaken in patients with brain tumours before surgical intervention [15]. Minimal research of decision making capacity after surgery exists, particularly in primary brain tumours. The research relating to decision making capacity following brain tumour removal often relates to brain metastases, as these patients often undergo multiple treatments and surgeries due to the recurrent nature of secondary brain tumours [36]. This lack of capacity assessment persists, despite evidence of cognitive changes following brain tumour surgery [37]. 

Capacity to make decisions relating to medical treatment is impaired in up to half of patients with malignant glioma [1]. These patients have difficulty in understanding the treatment situation, choices, and risks and benefits associated with the choices, and providing a rational reason for their decision. However, their ability to express a choice was comparable to controls [1]. Impaired decision making capacity is similarly common in patients with brain metastases [14], with up to 60% of patients with metastases assessed as having compromised capacity. Similar to patients with malignant glioma [1], patients with brain metastases had particular difficulty with the understanding and reasoning aspects of consent, and minimal difficulty expressing treatment choice. Thus, patients with brain tumours may be able to communicate their decision, but this does not necessarily reflect capacity to make that decision. If the patient’s decision is in line with that expected by the treating doctor, then a formal capacity assessment may not be initiated, or an assessment of their understanding of their decision may not be conducted [38]. 

Cognitive deficits can impact on a patient’s ability to provide informed consent for medical treatment [1]. This is especially relevant in the brain tumour population as there are often recurrences or progression, resulting in multiple treatments [39]. Consent to undergo surgical intervention or other medical treatment is only considered valid when the patient has capacity to consent to such intervention [7]. Informed consent involves the competent patient engaging with a medical practitioner to become fully informed about the proposed medical treatment, and making a decision regarding their treatment [40]. The process of obtaining informed consent involves a number of elements, including (1) discussing the medical issue and the decision to be made; (2) discussing the alternative to the recommended treatment; (3) discussing the benefits and risks of the treatment and the alternatives; (4) discussing patient concerns with the treatment; (5) assessing the patient’s understanding; and (6) eliciting the patient’s decision [38]. Despite these requirements for informed consent, primary care physicians and surgeons in general often fail to discuss one or more of the elements [38,41], even in high-risk surgeries [42]. Underpinning the notion of informed consent is the capacity to understand, evaluate, choose, and communicate a medical treatment decision, and this is frequently impaired in patients with brain tumours [1,14]. Consequently, capacity assessments are likely to be required for the majority of patients in this population. 

### 2.3. Return to Occupational Functioning

As cancer survivorship increases, quality of life and participation in occupational roles becomes more important [43]. The effects of brain tumours and their treatment on cognitive functioning can affect the individual’s ability to engage in work or schooling [44]. Cognitive changes can differentially affect patients’ ability to participate in work, depending on the type and cognitive demands of the work [45]. Insight into one’s ability to decide to return to work is important to consider for those in roles that demand the consistent ability to exercise judgement and insight [36]. In these roles, serious consequences may occur due to errors arising from even minor cognitive changes [46], and insight into the effects of these changes may be limited following brain tumour resection [45]. Thus, the capacity to decide to return to work may be affected in some patients post-brain tumour removal. 

### 2.4. Research Participation

In addition to decisions regarding treatment and other aspects of their lives, brain tumour patients often face the decision of whether or not to participate in research. Given a lack of effective treatments, patients are crucial for medical research, and medical research is crucial to improving survival rates in this population [47]. Additionally, participation in clinical trials is linked to better patient outcomes through improvements to the infrastructure of care provision, and through the process of how care is provided to patients [48]. However, patients may lack the capacity to consent to research participation. Patients with brain tumours frequently demonstrate impairments in their ability to consent to research participation and patients receiving steroid treatment and anticonvulsant medication have more difficulties with the components of informed consent, including understanding, abstract reasoning and evaluating information [49]. Thus, similar to capacity to consent to treatment [1,14], patients with brain tumours have a high likelihood of experiencing changes in their ability to understand and reason about all the relevant information on risk and benefits of participating, and appreciate the consequences of their decision to participate or not [49]. Importantly, capacity to consent to research (and treatment) is likely to decline with cognitive abilities in the more advanced stages of the disease process, which may be when participation is most sought after [49]. 

### 2.5. The Role of Caregivers

Caregivers play an important role in supporting the patient to make decisions, and to help inform the decision making capacity assessment. The involvement of caregivers in the decision making process is necessary as the management of brain tumours often occurs in the context of interpersonal relationships, with caregivers proving much of the day-to-day care to the patient [50]. Caregivers provide emotional support and contribute the relevant information needed to provide effective treatment [51]. Even when patient decision making capacity is not questioned, caregivers are heavily involved in treatment decision making [52]. Caregivers frequently serve as a conduit for information between the health care provider and the patient, obtaining information that can later be relayed and explained to the patient [52]. This role becomes especially important when the patient is experiencing cognitive changes that inhibit the processing and understanding of information. The relationship between the caregiver and the patient means that the caregiver is in a unique position to preempt information the patient may need to make their decision, and elicit this from the health care provider [52]. Caregivers also provide an avenue for the patient to discuss and consider their options, which can lead to greater confidence of the clinician that the patient has thoroughly considered the treatment options [52]. Caregivers are also better equipped than clinicians to identify when a patient’s decision is uncharacteristic or contradictory to previously expressed beliefs or attitudes, which could trigger a formal assessment of decision making capacity [12]. 

## 3. Capacity Assessment

At present, there is a lack of consensus on the most effective process for assessing capacity in brain tumour patients. However, there does seem to be agreement that cognitive changes are associated with difficulties in making decisions [2,15]. Neuropsychological assessment is considered to be the “gold standard” for assessing cognitive functioning and decision making in patients with a brain tumour, particularly as there is high heterogeneity in the cognitive profiles of these patients [53,54]. The capacity assessment process can identify potential barriers to a patient being able to make a decision, which can lead to remediation or alternative communication strategies to minimise restrictions on their decision-making powers [2,55]. Adapted decision making through the use of compensatory strategies will be greatly beneficial in patients with diminished capacity. Such strategies include simplifying and clarifying the specific decision to be made via pictures, simplified language, and limited options. Accurate assessment of cognitive functioning is important as it has implications for opinions formed and decisions made on the basis of this information, which can influence determinations of a patient’s ability to make decisions and also the support they receive. However, the classification of cognitive impairment has been inconsistent, with classification being based on impairment at test specific, domain specific, or global levels [53]. Test-specific impairment is classified on the basis of impaired performance on a single test (e.g., an individual working memory score), whereas domain specific impairment is classified by impaired performance on measures across a particular cognitive domain, such as attention [53]. Global level impairment is classified by impaired performance on a calculated composite score that spans different cognitive domains. Brain tumour patients are often impaired on at least one individual test of cognition, but less frequently at the domain specific and global levels [53,56]. Thus, the use of an individual test to conclude whether or not a patient is impaired will most likely be misleading, and assessment should include a variety of tests assessing a multitude of domains [53]. When risk factors for impaired decision making are identified, a brief yet tailed neuropsychological assessment combined with obtaining an understanding of the patient’s knowledge specific to the decision to be made will be essential in assessing decision making capacity [54]. 

### 3.1. The Role of Fatigue

Fatigue is an important a factor to consider when assessing patients post-brain tumour resection. Fatigue has been conceptualised as a psychobiological state manifested by a reduction in the efficiency of cognition, resulting from extended periods of activity that are cognitively demanding [57]. Fatigue is commonly experienced by patients following surgical resection of a brain tumour [58]. Fatigue in this population can be caused by a number of factors, with surgical treatment being one [59]. Other factors that can influence fatigue include the direct effect of the tumour, comorbid medical conditions (e.g., infection, anaemia), psychological disturbance such as anxiety and depression, and symptoms such as pain and disturbed sleep [59]. Fatigue can have a significant negative effect on cognitive processes [60], including goal-directed attention [61], sustained attention, and executive functioning, including planning, repetition of a response (i.e., perseveration) [62], and inhibiting an automatic response [63]. Given fatigue is a common complaint of brain tumour patients [64], the use of one test to infer impaired or intact cognitive skills is likely to be misleading. Thus, neuropsychological assessment of brain tumour patients must balance improved performance on brief assessments with the potential cost of reduced accuracy [65]. Brief neuropsychological assessments are tailored test batteries assessing a wide range of cognitive abilities, but take less time to administer than a standard battery. Standard batteries can take up to 8 hours to administer, which is almost intolerable for many patients post-brain tumour resection [66]. This is particularly important in the context of a capacity assessment where cognitive performance informs decisions on a patient’s ability to make decisions affecting their life. 

### 3.2. Cognitive Screening Tools

Given the need for brief assessment of cognition, alternatives to the more time-consuming neuropsychological battery strongly feature in research. Some researchers have focused their attention on an attempt to identify specific cognitive tests or screening tools that may be predictive of decisional incapacity. For example, Kerrigan et al. [15] have proposed a measure of semantic word fluency to be predictive of capacity to make treatment decisions in an intracranial tumour population. Triebel et al. [1] suggests verbal memory and semantic word fluency as predictors of deficits in the ability of patients to appreciate the consequences of their choice of treatment; understanding the treatment options, their risks and benefits; and providing a rationale for their choice. However, semantic word fluency and verbal memory are not the only cognitive skills that can impact decision making. In patients with brain metastases, processing speed and episodic memory have been found to be associated with decision reasoning abilities [67], and simple attention, processing speed, and executive functioning were associated with understanding of the decision and its components [68]. Thus, using one measure to assess one domain of cognition that is associated with decision making is unlikely to be predictive of a patient’s ability to make decisions, and unlikely to be sufficient to trigger a formal capacity assessment. 

Brief cognitive screening tools have also been suggested as time efficient tools in the assessment of decision capacity. Such tools have been used to support decisions regarding capacity and to identify patients at risk of impaired capacity in other populations, using screening tools such as the mini-mental state examination (MMSE) [69] and the Montreal cognitive assessment (MoCA) [70]. However, these measures are not sufficient for use in the determination of capacity. The MMSE was originally developed for assessing global cognitive decline, such as in Alzheimer’s dementia, depression, and delirium in an elderly population [71], and was not designed to assess deterioration in cognitive functioning in patients with other neurological conditions [72]. Similarly, the MoCA was developed to detect mild cognitive impairment and Alzheimer’s dementia [73]. Notably, the MMSE has been shown to have limited utility in discriminating those with and without capacity in the Alzheimer’s dementia population [74], non-specified dementia, psychiatric, and acquired brain injury populations [75]. Not only has the MMSE been shown to have poor predictive value for capacity [76], it and the MoCA have been shown to have limited utility in the brain tumour population. Both the MMSE and the MoCA have been found to have poor sensitivity to cognitive deficits in patients with brain tumours [37,77,78], suggesting these screening tools would contribute little to the cognitive assessment component of capacity assessment, particularly in the brain tumour population. Therefore, the use of a cognitive screening tool at any point in the assessment of decision making ability in this population is likely to be misleading. 

### 3.3. Decision Specific Assessment

Another central component of capacity assessments is the evaluation of the skills and knowledge related to the specific decision to be made [2]. Capacity assessments are decision specific, and not reflective of global decision making ability, or even other decisions within the same domain (e.g., medical decisions) [79]. Thus, assessments of decision making capacity must include assessment of the specific knowledge relating to the decision, which can be achieved through structured interview, semi-structured interviews, question sets, vignette-based assessments, or a combination of these [2]. Neuropsychologists use a combination of formal neuropsychological assessment and interview information to assess capacity, while approaching each case and capacity question with flexibility [54]. Structured interviews are tools to support clinical judgment on capacity, not as a replacement for this process, as strictly structured assessments are unlikely to detect subtle individual nuances [54]. The semi-structured interviews and question sets focus on asking questions to elicit an understanding of the patient’s knowledge relating to the decision in question, as well as questions tapping other factors that have the potential to influence decision making, such as perceptions of quality of life [2]. Vignettes can take this further by eliciting information about a patient’s preferences, such as in making a health related decision [80]. Vignettes involve providing the patient with an imaginary situation in which they decide on a proposed treatment, and an interview is used to assess their understanding of the relevant issues [81]. The combination of formal neuropsychological assessment and interview can be invaluable in assessing the functional consequences of cognitive changes on the process of decision making, including reasoning, understanding, and communicating [54]. Such an approach is likely to present a more robust determination of capacity than either approach used in isolation [2]. 

### 3.4. The Role of the Multidisciplinary Team

A multidisciplinary approach to capacity assessment is likely to provide the most accurate assessment. Although this may not be practical or possible in most outpatient settings, utilising the expertise of other professionals can inform the capacity assessment [82]. For instance, solicitors may be needed to help determine capacity to make legal decisions when matters are complex. Of particular relevance to medical decisions, the treating doctor or consultant doctor involved in the assessment process can help to ensure that the patient’s understanding of the medical treatment is sufficient [41]. Neuropsychologists conducting capacity assessments need to be acutely aware of their scope of practice, and enlist the assistance of other professionals to obtain a clear understanding of the patient’s capacity. Additionally, functional assessments are useful in determining a patient’s understanding of their decisions by providing information of any risks (e.g., risk of falling) when released home [83]. A functional assessment completed by an occupational therapist would provide valuable information relating to a patient’s function and needs, and regarding what modifications can be made or supports utilised in order to maximise participation in their own care [83]. This information can be useful when considering or assessing the patient’s level of insight into their limitations and needs. 

## 4. Cognition in Decision Making

In practice, decision making is a process by which an individual makes predictions about which option or course of action among alternatives is most likely to achieve a goal [84]. Capacity has been conceptualised in terms of specific competencies, rather than capacity as a unitary concept, although it has been argued that there are common cognitive processes that underlie a person’s ability to make an informed decision [2]. It is important to note that neuropsychological assessments were not specifically designed to assess decision making capacity [2], but are useful as decision making requires abilities that have a neurological basis [85]. A number of cognitive abilities play a role in the decision making process, including executive functioning [86], attention [87], language [88], working memory [89], and reasoning ability [90]. Emotions have also been found to play a strong role in decision making [91] and the impact of emotions on decision making becomes especially important with the brain tumour population, as emotional dysregulation, disinhibition, and changes in affect are a common consequence [92]. Cognitive changes can occur due to the effects of the tumour itself, as a consequence of surgery, or a combination of these [28], and these changes can persist following tumour resection. The proliferation of brain tumours into surrounding healthy brain tissue can cause cell death via compression of healthy tissues, leading to the restriction of oxygen and nutrients from reaching normal cells [93]. Neurodegeneration of healthy brain cells can also occur through neurotransmitter toxicity, which is caused by impaired neurotransmitter uptake by glial cells in glioma tumours (i.e., glutamate excitotoxicity) [94]. Cognitive changes can also occur as a result of tumour resection due to damage to the surrounding healthy tissue [20], although these deficits are often transient [28]. Following tumour resection, cognitive functioning is frequently improved, although many patients continue to experience a wide range of changes [17,20]. It is difficult for the effects of tumour resection surgery to be differentiated from the effects of the tumour itself, as the cognitive changes are likely due to a combination of factors. 

### 4.1. Executive Functions

Executive functions encompass a set of mental skills needed to achieve a goal, such as planning, organising, focusing attention, initiating or inhibiting a response [95]. Executive functions play an important role in making decisions [86]. Underlying successful decision making are flexibility and adaptability in identifying the available choices, selection or inhibition of choices that are immediate and may or may not be in the person’s best interest, evaluating the choices in terms of probabilities and value, and predicting the impact of a choice on the overall goals [86]. Executive functions are widely held to be supported by the frontal lobes, with the prefrontal cortex being responsible for the selection and execution of actions, and posterior brain areas supplying the necessary information in order to select the appropriate action and the method for executing the chosen action [96]. The prefrontal cortices have been implicated in flexible decision making [90] and damage affecting the inferior medial prefrontal cortex has been linked with abnormal decision making, particularly in a social context [97]. Salver and Damasio [97] reported a patient with damage to this area and, although core capacities for decision making were intact, such as knowledge base, considering the future consequences of a decision, and predicting a likely outcome, the patient had difficulty choosing or implementing a selected option. The difficulties in selecting an option were thought to be linked to impaired calculation of a selected action’s value [98], which can lead to an individual choosing more exploitative choices [99]. The inferior lateral prefrontal cortex has been associated with self-control, such as that needed for inhibiting inappropriate or incorrect responses, and delayed gratification [100]. In brain tumour patients in particular, the right inferior lateral frontal region has been implicated in inhibition and strategy implementation [101]. Disturbance in these functions have implications for decision making. 

Changes in executive functions resulting in impulsivity have been implicated in impaired decision making [102]. Impulsivity is a common consequence of damage to the orbitofrontal cortex, and involves behaviours that are premature, inappropriate, or inadequately planned, and typically result in undesirable outcomes [103]. Poor decision making in daily life is a common manifestation of damage to the orbitofrontal cortex, and can be conceptualised as increased engagement in risk-taking behaviour [104]. Further, impulsivity impairs an individual’s ability to make appropriate decisions in an uncertain context and to change a behaviour in the context of new information [102,104]. This is relevant for patients making decisions regarding treatment in the light of new information about their illness. Similarly, under conditions of uncertainty, mental flexibility is needed for evaluating available information in order to decide how to act, a process the orbitofrontal cortex may be critical for [103]. Executive functions are required in medical decision making in terms of integrating past experiences and current action into a personal framework to facilitate understanding of the information provided [68]. Therefore, adequate executive functions appear essential for the evaluation and understanding of information needed to make an informed decision. 

### 4.2. Attention

Attention supports the role of executive functioning in making decisions. The mental flexibility required to allocate cognitive resources to achieve goal-directed behaviour is facilitated by cognitive control, an executive function that involves the prioritisation of information needed to achieve that goal [87]. Attention is one mechanism that influences cognitive control by biasing information processing to support goal achievement [87]. Attention can be viewed as a filtering mechanism that selects which information to process more deeply [105], and the task difficulty determines the amount of cognitive resources needed to process the information [106]. The more difficult a task, the more attentional resources it requires. Dysregulated attention may have negative implications for making decisions in the face of large amounts of information. Complex decisions like deciding on further medical treatment for brain tumours, require conscious deliberation and allocation of a high level of attentional resources [106]. Attention is frequently affected in patients with brain tumours both before and after surgical resection [21], and changes in divided and focused attention have been linked to lesions to the prefrontal cortex [107]. Attention has been found to be associated with comprehending the information provided in order to make decisions relating to medical treatment [68]. The ability to attend to relevant information is imperative for making informed decisions, particularly those regarding medical care. 

### 4.3. Working Memory

Working memory refers to the ability to hold, process and use information and manipulate it in some way to achieve a goal [108]. Working memory is imperative for decision making as it is needed to hold in mind information about the value of the expected outcome of a decision [75]. Increases in the amount of information an individual needs to hold in mind (i.e., working memory load) impairs decision making [109,110,111,112]. Working memory is involved in continuous information updating based on past experiences, and increased working memory load can lead to making choices based on immediately foreseeable outcomes [110]. High working memory capacity is associated with the ability to engage with rational or logical processing, rather than experiential or intuitive decisions, and is likely mediated by better attentional control [111]. Working memory changes can lead to poorer abstract or rational reasoning and cognitive inflexibility and ignoring alternative choices, which results in making decisions based on biases, rather than on the available, up to date, and relevant information [111]. Changes in working memory can have negative implications for decision making in patients who have undergone surgery to remove a brain tumour. 

The somatic marker hypothesis [113] proposes that bodily feelings associated with emotions (i.e., affective somatic states) are associated with previous decisions, which then guide future decisions. That is, decisions with poor outcomes are associated with negative states and positive outcomes are associated with more positive states [114]. Ongoing choices reinforce these associations and over time the emotional reactions precede the decisions and influence which choice is made [114]. Working memory contributes to the development and utilisation of these somatic markers, and difficulties in this obstruct the cognitive processes required to cement the anticipatory affective reactions [115]. Changes in working memory thereby contribute to poor decision making. The prefrontal cortex plays a core role in working memory [116], and the orbitofrontal cortex in particular is implicated in everyday decision making [89]. 

### 4.4. Memory

It makes intuitive sense that memory plays a role in decision making. Almost all decisions will require the ability to recall prior actions and their associated emotional responses in individual events at a particular place and point in time [117]. Semantic memory that comprises memory for facts and general knowledge has been proposed to play a role in decisions that require the recall and use of background knowledge to help inform decision making [118]. Episodic memory for autobiographical events contributes to decision making through the appraisal of relevant past experiences to make memory-based judgments [118]. The medial prefrontal cortex facilitates memory in decision making by mapping context and events onto relevant actions, and facilitating recall of these actions and their associated emotional responses in guiding decisions [117]. The dorsolateral prefrontal cortex supports this process through its role in evaluating information and selecting an appropriate action response [119]. Memory changes in patients following brain tumour resection may affect decision making, particularly for medical decisions in individuals who have had previous treatment with associated memories to draw from. 

Verbal memory in particular has been implicated in decision making [1]. Understanding and reasoning with the relevant information is of particular importance in determining a patient’s decision making capacity [36], and changes in verbal memory can influence these skills [1,69]. Encoding, storing and recalling of verbal information after it is presented [120] are aspects of verbal memory. Difficulties in any of these stages of verbal memory can impair patients’ understanding of the information [1]. Further, reasoning with the information provided requires a functioning verbal memory, as the patient needs to keep verbal information in mind to compare the relative risks and benefits associated with each decision, and to use that information to make a decision [56]. Patients with poor verbal memory can experience difficulties with decision making due to having less information to consider when reasoning and appreciating the significance of the information [1]. The temporal areas have been implicated in verbal memory and tumours in these areas may compromise verbal memory and have implications for decision making capacity [121].

### 4.5. Language

Language is an obvious skill required for decision making as it is necessary to understand information provided to inform decisions. As mentioned, decision making capacity involves (i) understanding and appreciating the significance of relevant information for a specific decision; (ii) reasoning with the information; and (iii) expressing a decision [56]. When considering the process for assessing capacity, namely neuropsychological assessment and qualitative interviews [55], language is imperative. Understanding of language is essential in neuropsychological assessment, and without the ability to process verbal information, patients cannot be expected to effectively complete an assessment [122]. Similarly, during the assessment of the specific knowledge relating to the decision to be made, proficient use of language past a basic yes or no response is required [123]. Reasoning and problem solving is commonly negatively affected in those with acquired language disorders (e.g., aphasia [124]), and these skills represent a main component of making informed decisions [88]. The ability to express thoughts, feelings and desires can be disturbed by deficits in the executive control of language, caused by damage to the prefrontal cortex. This can be due to reduced initiation or selection of language [125,126] or even due to an inability to inhibit verbal responses [127]. The intersection between language and executive functions is crucial for decision making and expression of a personal choice. Many cognitive functions can be intact in patients with communication disorders, meaning they should not be assumed to lack decision making capacity due to an inability to communicate that capacity through conversation [89]. 

Language disorders affecting a person’s ability to understand or use language (i.e., aphasia) can affect verbal comprehension, expression, written expression and reading comprehension to varying degrees [124]. Language changes can occur in up to one third to one half of patients whom have undergone surgical resection of left hemisphere tumours [128]. Often, language changes are present prior to surgical resection and are frequently improved following resection, although language skills can be worsened following surgery [128,129]. Difficulties with naming known objects (i.e., anomia) is the most common type of language disorder following surgical resection in the left temporal lobe, and is often mild [128]. However, more infiltrative tumours (e.g., glioblastomas) tend to produce more severe language deficits [128]. Despite the level of language impairment present, decision making incapacity is not inherent in aphasia, although patients may be limited in their ability to fully partake in discussion regarding a specific decision to be made [130]. This point is illustrated using two patients presenting with differing language abilities. Table 1 presents the neuropsychological profiles of two patients assessed post-resection of their brain tumours. Box 1 describes the comparison of the two patients in terms of their neuropsychological profile, and the implications of their cognitive functioning on decision making capacity, particularly the role of language. Box 2 provides suggested clinical guidelines for assessment of capacity for decision making and the core domains involved in decision making. Figure 1 presents a flow chart representing the process for assessing decision making capacity, adapted from Sullivan [2]. 

Box 1Case Illustration.Table 1 presents the neuropsychological profile of two patients. Patient JD, a 64 year-old female underwent a stereotactic craniotomy and resection of a left temporal glioma, 17 months prior to testing. Patient KG, a 73 year-old male was tested within two weeks of undergoing a stereotactic craniotomy to remove a right frontotemporal meningioma. JD and KG were relatively well matched for intellectual functioning, and presented with different neuropsychological profiles. When considered in the context of decision making capacity, both JD and KG had indicators of potential difficulties in this process. Capacity assessments include clinical interview with the patient, which is heavily reliant on language abilities [123]. JD had reduced spontaneous speech and naming difficulties, which could affect others’ perception of her ability to make informed decisions. JD’s receptive language skills were intact, as evidenced by good ability to understand and follow instructions and high average reading ability. JD may be capable of understanding her options, and their respective consequences, but have difficulty may effectively articulating her reasoning and communicating a decision. If clinical interview alone was used to assess decision making capacity, it is likely that JD would be identified as lacking capacity due to her expressive language difficulties. JD’s poor verbal memory and executive dysfunction would have a greater impact on her ability to effectively make decisions; however, this would unlikely be identified through interview alone. By contrast, KG presented as articulate, with no difficulties in spontaneous speech, naming, or word comprehension. Through interview alone, he would likely have no difficulty communicating a choice, and may be perceived as having decision making capacity. However, KG had difficulties with verbal and visual memory and some aspects of executive function. The verbal and visual memory impairments could limit his ability to retain information relating to decisions to be made, as this information is often provided verbally. His retention of this information may not be improved even if it was presented in writing (i.e., visually). KG presents with disinhibition within a semantic context, which could lead to impulsive decision making, via an impaired ability to inhibit the selection of inappropriate choice. That is, KG may choose options that are immediate and not in his best interest, he may not effectively evaluate his choices, and may be unable to predict the consequences of his choice. The differing profiles of abilities highlight the importance of assessing a wide range of cognitive abilities in assessing decision making capacity, and not relying on interview alone. Both patients present with deficits in executive functioning; however, JD would be more likely to be identified as potentially having difficulties with the decision making process due to her expressive language difficulties. Importantly, although most likely to be more easily identified as a potential barrier to decision making capacity, expressive language deficits do not equate to decision making incapacity. Further, executive dysfunction may be more suggestive of decision making difficulties, but this would not be evident in a clinical interview alone. Of particular note is that only JD was identified to be cognitively impaired using the MoCA screening tool. KG was intact on the MoCA, despite having significant difficulties with verbal learning and memory, and executive functions. This provides further support to a growing body of evidence that the MoCA is not sensitive enough to detect the specific and/or focal deficits experienced by brain tumour patients [37,78,79]. 

Box 2Clinical guidelines for the assessment of the capacity for decision making.**Neuropsychological Assessment** (ideally pre-post brain tumour resection)-**Executive functions:** flexibility and adaptability in identifying the available choices, selection or inhibition of choices that are immediate, evaluating the choices in terms of probabilities and value, and predicting the impact of a choice on the overall goals [86].-**Language:** ability to understand and appreciate the significance of relevant information for a specific decision, and express a decision [56].-**Memory:** recall of previous actions and their associated emotional responses in individual events at a particular place and point in time [117].-**Attention**: conscious deliberation and allocation of a high level of attentional resources [106].**Decision specific assessment of knowledge and skills related to each issue** (e.g., medical treatment, return to work or other activities, managing finances, etc.)

## 5. Emotion in Decision Making

Emotions play a large role in influencing judgements and choice. Affective disposition can influence information processing style, with processing information relating to decisions becoming more extensive as negative emotion increases, leading to focusing on one attribute at a time [144]. The emotional value attributed to a decision (i.e., emotional valence) affects predictions about future consequences, with fearful people tending to make more pessimistic judgement about future events [145]. This is particularly relevant when considering the emotions experienced by patients following brain tumour removal, especially those whom may have more treatment decisions ahead of them. Emotions can influence memories, with memories congruent with current mood states being more easily recalled than memories that are incongruent with current mood [146]. Similarly, the somatic marker hypothesis [114] proposes that decisions are guided by the emotional states associated with previous decisions. Over time, decisions to be made are increasingly associated with either positive or negative emotions, as determined by previous similar choice outcomes, which then influence the decision to be made [115]. Patients experiencing tumour recurrence or metastases whom have had previous treatment may be particularly vulnerable to these associated emotional states when making treatment decisions, particularly if that experience was not positive. 

The influence of emotions on choice is important in the brain tumour population, as these patients can experience changes in processing emotional information, and emotional dysregulation and disinhibition [92]. Difficulties in processing emotional information can impair decision making, especially decisions of a personal or social nature, despite otherwise intact intellectual functioning [92]. The inferior medial prefrontal cortex, including the orbitofrontal cortex, has been implicated in the processing of emotional or bodily signals [92,147,148]. Lesions to this region predominantly affect personal and social decisions, including life planning, and choosing friends and partners [149]. Emotion dysregulation can be a consequence of brain tumours, especially when the lateral prefrontal cortex is involved [150], and difficulties regulating emotions can lead to a tendency to make riskier decisions [151,152]. Thus, the ability to control emotional responses can promote decision making that is goal-directed [153]. This means that brain tumours patients experiencing changes in emotional control could be vulnerable to risk taking, which has implications for treatment decisions. 

## 6. Conclusions

Decision making capacity is underpinned by a number of cognitive functions. Changes in cognitive functioning are common in patients following brain tumour resection, and these changes have implications for their ability to make decisions [1]. Unfortunately, capacity assessments are rarely conducted in this population [7,15], despite the plethora of research to indicate capacity to make decisions about a variety of issues is often impaired [6,7,14]. Brain tumour patients can experience these changes even after surgical resection of the tumour, suggesting their ability to make decisions regarding, and consent to, adjuvant therapies or additional treatments may be compromised [13]. This is particularly relevant in cases of higher grade tumours or metastatic tumours, as these tend to have an aggressive and progressive course, which can lead to rapid deterioration in cognitive functioning, and potentially, decision making capacity [1]. Neuropsychological assessment plays an imperative role in the assessment of decision making capacity and should entail evaluation of a wide range of cognitive skills [2]. Given the role of many cognitive functions in decision making, and the natural variability in cognitive strengths and weaknesses, the use of one measure or assessing only one cognitive domain to infer capacity or incapacity is likely to be misleading [54]. This is particularly important when considering that fatigue is a common complaint within the brain tumour population that affects test performance [59]. Further, the use of brief cognitive screening tools such as the MMSE and MoCA have been shown to be insensitive to the specific and focal deficits experienced by brain tumour patients [37,78,79], further reinforcing the need for more extensive albeit brief neuropsychological assessment. 

Capacity to make decisions is restricted to a specific decision to be made at a specific point in time, and not a general decision making capacity [4,5]. Neuropsychological assessment can provide an understanding of the patients’ cognitive functioning and any likely impact on decision making ability, and assessments or interviews targeting decision-specific knowledge are required for each decision to be made [2]. Additionally, assessments of patient physical function is important for capacity assessments as they facilitate an understanding of a patient’s level of insight into the modifications or supports needed to maximise participation in their own care [83]. Accordingly, decision making capacity assessments are likely to be most effective when multiple disciplines are utilised, particularly when assessing specific knowledge relating to a decision that is outside the scope of practice of the neuropsychologist [84]. When assessment of neuropsychological functioning and decision specific knowledge reveal deficits in decision making ability, the patient must be supported to make that particular decision. Neuropsychological assessment can be used to identify areas for cognitive remediation for skills identified as impaired, which may help to improve the patient’s capacity to make decisions [2]. Understanding a patient’s cognitive strengths and weaknesses will also allow for compensatory strategies to be implemented to support decision making, such as providing information in writing or repeating information over time for memory impaired patients, or delivering information slowly to patients with reduced processing speed. Adapted decision making through the use of these compensatory strategies will also be greatly beneficial in patients with diminished capacity, and most useful in settings with limited resources. Additionally, caregivers are vital in supporting the patient and facilitating decision making [51,52], and monitoring for signs that their decision making capacity may be compromised, thereby triggering a capacity assessment [12]. The limitations of many clinical settings in terms of the staffing and resources needed to support patients means utilising the support of caregivers help to make decisions will be invaluable. 

Overall, when a patient’s decision making capacity is questioned, efforts must be made to ensure the course of action taken is the least restrictive in terms of patient rights [2,43]. Neuropsychological assessment provides an avenue to determine the cognitive skills impacting on the patient’s ability to make decisions and potentially inform rehabilitation strategies to overcome barriers and increase decision making capacity [2]. A capacity decision made by the treating medical team that is supported by formal cognitive assessment and other objective evidence of capacity provides protection in the event of a dispute [7]. Therefore, neuropsychological assessment as part of the process of assessing capacity is likely to result in better outcomes for both patient and the treating medical team, and should be considered as a matter of course when working with patients with brain tumours.

## Figures and Tables

**Figure 1 brainsci-07-00122-f001:**
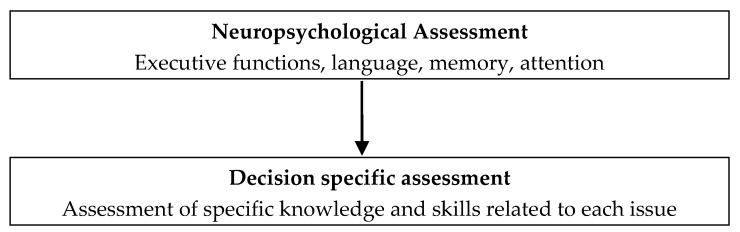
Flow chart of the approach to assess decision making capacity.

**Table 1 brainsci-07-00122-t001:** Comparison of the neuropsychological profiles of two patients.

DOMAIN/TEST	Patient JD	Patient KG
COGNITIVE SCREENING TEST	Raw Score (Percentile)	
MoCA [57]	**21/30 (Impaired)**	27/30 (Intact)
**INTELLECTUAL FUNCTIONING**		
Fluid intelligence (RAPM ^1^ [131])	4 (23)	7 (72)
Premorbid estimate (NART ^2^ [132])	FSIQ = 111	FSIQ = 113
**MEMORY**		
Working memory (Digit Span, WAIS-III ^3^ [133])	13 (25)	16 (63)
Verbal memory—learning (RAVLT ^4^ [134])	**27 (<1)**	26 (~23)
Verbal memory—recall (RAVLT)	**2 (<1)**	**2 (1)**
Verbal memory—delayed (RAVLT)	**2 (<1)**	3 (16)
Visual memory—recall (RCFT ^5^ [135])	**14.5 (<10)**	**10 (<10)**
Visual memory—delayed (RCFT)	20.5 (17)	**11.5 (<10)**
**VISUO-CONSTRUCTIONAL**		
Visuo-constructional (RCFT)	36 (>99)	**25.5 (<10)**
**LANGUAGE**		
Spontaneous speech (cookie thief [136])	**101 words/min**	129 words/min
Naming (graded naming test [137])	**6 (<1)**	18 (25-50)
Concrete word comprehension (synonyms [138])	17 (25)	23 (75)
Abstract word comprehension (Synonyms )	20 (37)	23 (75)
**EXECUTIVE FUNCTIONS**		
Verbal fluency—phonemic (F [139])	**2 (<1)**	11 (25)
Verbal fluency—phonemic (S)	**8 (5)**	**4 (2)**
Verbal fluency—semantic (Animals)	**10 (5)**	13 (16–25)
Inhibition (Stroop [140])	**66 (14)**	72 (16)
Verbal initiation reaction time (Hayling [141])	Moderate Average	Average
Verbal suppression errors (Hayling)	**Impaired**	**Impaired**
Trail Making B ([142] completion time)	**116′′ (<1)**	116′′ (50)
Trail Making A (completion time)	32′′ (50)	45′′ (50)
**ATTENTION**		
Selective auditory attention (TEA^ 6^ [143])	9 (50)	10 (75)
Selective visual attention (TEA)	4.4′′ (10–25)	5.5′′ (10–25)
Dual visual and auditory attention (TEA)	5.95′′ (25–50)	6.05′′ (25–50)

Bold indicates impaired performance. ^1^ Raven’s Advanced Progressive Matrices; ^2^ National Adult Reading Test; ^3^ Wechsler Adult Intelligence Test, Third Edition, ^4^ Rey Auditory Verbal Learning Test; ^5^ Rey Complex Figure Test; ^6^ Test of Everyday Attention.

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
