# Peer review of "Changes in Cognition and Decision Making Capacity Following Brain Tumour Resection: Illustrated with Two Cases"

_brainsci, 2017, doi:10.3390/brainsci7100122_

Round 1

Reviewer 1 Report

The authors present an extensive  literature review about the changes in decision making and decisional capacity that might arise as a consequence of having a brain tumour or undergoing brain surgery for tumour removal. The importance of decision making assessment is, at the end of the paper, illustrated by means of the description of two clinical cases.

I think the manuscript is well written and articulated, covering a wide range of clinical literature oriented to brain tumours cognitive assessment and covers an interesting and often not well considered issue regarding the clinical management of patients with brain tumours.

Herewith a series of comments that might improve the clarity of the manuscript in my opinion

1)      The paper focuses on the concept of “Capacity” which is clearly conceptualised and defined in the paper at line 35. However, although the concept is central throughout the paper, the manuscript seem to lack a more “operationalised” description of it. In other words: how is capacity formally evaluated? Are there any guidelines?  It is clear that a proper assessment of the capacity to make a decision is critical at different stages of the illness for brain tumour patients, but HOW this construct is operationalised in terms of cognitive functions? Language, memory, attention and executive functions are clearly important, but this is quite like claiming that the integrity of almost all higher cognitive functions is critical for the preservation of the capacity of making a proper decision. Which of these function plays a more critical role ? (executive functions in my opinion, followed by language and memory). Possibly a box summarizing the cognitive abilities critical for making a proper decision could be useful (more exhaustive than Box2).

2)      Another related issue: the manuscript would benefit from a clearer discussion of which tests are more appropriate to properly evaluate the capacity of patients to make a decision. This is only generally covered by Box 2. I think that Box two should be greatly expanded and the manuscript should cover a more detailed discussion of which of the most commonly used neuropsychological tools are more likely to be appropriate and sensitive in evaluating capacity in brain tumour patients, which are a particular category of brain damaged patients, in that they often show milder and more subtle cognitive impairments with respect to other neurological populations. This is partly done by authors by underlining that general screening tools such as MoCA or MMSE are inadequate, but I think that a list of specific tests (instead of just a list of domains) would be more useful, together with a brief description of why these are more likely to be suitable to evaluate capacity. This could be of great operational value for clinical neuropsychologists working with brain tumour patients.

3)      Brain tumours are a quite heterogeneous category of intracranial expansive lesions. They generally include High Grade Gliomas, Low Grade Gliomas, Meningiomas, Brain Metastases and Lymphomas (just to list the most common). I think the paper would benefit from a discussion and a brief summary of the main features of these different tumours, since they have quite different cognitive impact, and they are differentially impacted by surgery. This is only generally done by authors between lines 68 and 78, but I think that this would deserve a separate paragraph. For instance while High Grade Gliomas (HGG) tend to show cognitive impairment already before surgery (e.g. Kayl & Meyers, 2003; Noll, Sullaway, Ziu et al, 2015), Low Grade Glioma (LGG) patients tend to not show any cognitive deficit before surgery (e.g. Walker, Kaye, 2003; Desmurget, Bonnetblanc, Duffau, 2007). However after surgery LGG patients generally experience a significant detrimental effect of surgery over cognitive functions (e.g. Sheibel et al, 1996; Duffau et al, 2003, Campanella et al, 2015, Campanella et al 2017), but these deficits are generally recovered within few months. Meningiomas in general, show little cognitive impact in general and little effect of surgery (Tucha et al, 2001, 2003). Since the paper focuses on changes of capacity following brain tumours and tumours resection, considering different tumours together might be misleading, since different tumours are likely to differentially affect capacity of patients at different moments throughout the history of their illness.

4)      Another related issue regards the confusion between the effects of tumour itself and those related to surgery. Throughout the paper the authors describe interchangeably the cognitive deficits of brain tumours and those related to the surgery, which in my opinion should be treated separately. In the title itself it is suggested that the focus will be on those effects related to tumour resection, but most of the effects described relate to tumour itself, and the distinction is never made clear.

MINOR POINTS:

1)      At line 31: “…requires that an individual, including those with a brain tumour patients, “: I think the sentence has an error (remove the word “patients”)

2)      Line 62: it is stated that capacity assessment should be initiated any time a patient is displaying a risk factor for impaired decision making, and among the risk factors “having a brain tumour” is included. This basically means that capacity assessment should be routinely done when evaluating ANY brain tumour patient, which may be excessive.

3)      Lines 421-425: I think there is an error in the references. When ref 113 is indicated, I think authors are really referring to ref 114. Am I correct? Indeed I wonder whether ref 113 is appropriate at all.

4)      Table 1 is a bit odd to “decipher”: for a better reading, impaired scores should be highlighted somehow, and also cognitive domains should be more clearly separated. Moreover it lacks an explanation of the acronyms used and the respective references should be provided for the tests used.

5)      Finally: I think that case 2 (KG) might not be the best case to provide. While I completely agree that MoCA or MMSE are inadequate to provide a complete and clear cognitive picture in brain tumour patients, in the case of KG MoCA fails to show cognitive impairment…but in general KG seem to show VERY mild and limited cognitive impairments, and to my opinion the scores in executive tests are unlikely to suggest a real possible impairment in capacity and decision making.

Author Response

Reviewer #1

This reviewer thought our article may be of interest for the Brain Sciences readership and detailed several issues to address:

1)      The paper focuses on the concept of “Capacity” which is clearly conceptualised and defined in the paper at line 35. However, although the concept is central throughout the paper, the manuscript seem to lack a more “operationalised” description of it. In other words: how is capacity formally evaluated? Are there any guidelines?  It is clear that a proper assessment of the capacity to make a decision is critical at different stages of the illness for brain tumour patients, but HOW this construct is operationalised in terms of cognitive functions? Language, memory, attention and executive functions are clearly important, but this is quite like claiming that the integrity of almost all higher cognitive functions is critical for the preservation of the capacity of making a proper decision. Which of these function plays a more critical role ? (executive functions in my opinion, followed by language and memory). Possibly a box summarizing the cognitive abilities critical for making a proper decision could be useful (more exhaustive than Box2).

            * There is no universally accepted construct of capacity in terms of cognitive functions.  We present information on the roles various cognitive functions play in decision making to highlight that decision making involves many cognitive functions and a complex interplay between them.  Box 2 has been elaborated to include the likely four core functions necessary for decision making, in the authors’ opinions.  As with all neuropsychological testing, there is a level of clinical judgement required for interpreting the influence of an individual patient’s cognitive weaknesses on their decision making ability. 

2)      Another related issue: the manuscript would benefit from a clearer discussion of which tests are more appropriate to properly evaluate the capacity of patients to make a decision. This is only generally covered by Box 2. I think that Box two should be greatly expanded and the manuscript should cover a more detailed discussion of which of the most commonly used neuropsychological tools are more likely to be appropriate and sensitive in evaluating capacity in brain tumour patients, which are a particular category of brain damaged patients, in that they often show milder and more subtle cognitive impairments with respect to other neurological populations. This is partly done by authors by underlining that general screening tools such as MoCA or MMSE are inadequate, but I think that a list of specific tests (instead of just a list of domains) would be more useful, together with a brief description of why these are more likely to be suitable to evaluate capacity. This could be of great operational value for clinical neuropsychologists working with brain tumour patients.

            * As reviewer 1 points out, the cognitive deficits seen in brain tumour patients are often mild and focal in nature, meaning appropriate selection of specific neuropsychological tests is essential.  Unlike Reviewer 1, the authors believe listing or recommending which neuropsychological assessments would be most useful is inappropriate for this paper.  The authors are strongly of the view that discussing the domains most associated with decision making is most appropriate (and sufficient), as this allows clinicians to select a test with normative data that best fits the patient, especially if region specific test adaptations and norms are available (e.g. Australian adaptations for tests developed in the United States).  

3)      Brain tumours are a quite heterogeneous category of intracranial expansive lesions. They generally include High Grade Gliomas, Low Grade Gliomas, Meningiomas, Brain Metastases and Lymphomas (just to list the most common). I think the paper would benefit from a discussion and a brief summary of the main features of these different tumours, since they have quite different cognitive impact, and they are differentially impacted by surgery. This is only generally done by authors between lines 68 and 78, but I think that this would deserve a separate paragraph. For instance while High Grade Gliomas (HGG) tend to show cognitive impairment already before surgery (e.g. Kayl & Meyers, 2003; Noll, Sullaway, Ziu et al, 2015), Low Grade Glioma (LGG) patients tend to not show any cognitive deficit before surgery (e.g. Walker, Kaye, 2003; Desmurget, Bonnetblanc, Duffau, 2007). However after surgery LGG patients generally experience a significant detrimental effect of surgery over cognitive functions (e.g. Sheibel et al, 1996; Duffau et al, 2003, Campanella et al, 2015, Campanella et al 2017), but these deficits are generally recovered within few months. Meningiomas in general, show little cognitive impact in general and little effect of surgery (Tucha et al, 2001, 2003). Since the paper focuses on changes of capacity following brain tumours and tumours resection, considering different tumours together might be misleading, since different tumours are likely to differentially affect capacity of patients at different moments throughout the history of their illness.

            * We agree with this reviewer and appreciate the suggestion. We have now added additional information on the differing cognitive functioning before and after surgery in meningioma and glioma patients (from line 84 in the revised manuscript). 

4)      Another related issue regards the confusion between the effects of tumour itself and those related to surgery. Throughout the paper the authors describe interchangeably the cognitive deficits of brain tumours and those related to the surgery, which in my opinion should be treated separately. In the title itself it is suggested that the focus will be on those effects related to tumour resection, but most of the effects described relate to tumour itself, and the distinction is never made clear.

            * We have now clarified (from line 435 in the revised manuscript) that the cognitive changes observed following surgery are not able to be clearly attributed to the effects of the tumour itself or the surgery.  Cognitive changes are likely the combination of both surgery and tumour factors, including brain cell death as a result of compression of healthy tissues by the tumour (Jain, Martin, & Stylianopoulos, 2014) and glutamate excitotoxicity (Lee et al., 2011).  Importantly, the paper is simply addressing the cognitive changes that are present after surgery, without examining in depth the relative contribution of the factors affecting cognition. 

MINOR POINTS:

1)      At line 31: “…requires that an individual, including those with a brain tumour patients, “: I think the sentence has an error (remove the word “patients”)

            * This error as now been corrected. 

2)      Line 62: it is stated that capacity assessment should be initiated any time a patient is displaying a risk factor for impaired decision making, and among the risk factors “having a brain tumour” is included. This basically means that capacity assessment should be routinely done when evaluating ANY brain tumour patient, which may be excessive.

            * This paragraph has now been amended to clarify that it is not the recommendation of the authors that capacity assessments be conducted for every patient with a brain tumour, but capacity assessments should be considered when a brain tumour patient shows risks associated with decision making compromise.  Having a brain tumour is one such risk, as is the patient expressing a decision that is contrary to what they may have been expected to choose.  Additionally, the authors have included the use of pre- and post-surgery neuropsychological assessment as a means to help identify cognitive risk factors for impaired decision making, as well as the role of family members in helping to identify risks.

3)      Lines 421-425: I think there is an error in the references. When ref 113 is indicated, I think authors are really referring to ref 114. Am I correct? Indeed I wonder whether ref 113 is appropriate at all.

            * This error has now been corrected. 

4)      Table 1 is a bit odd to “decipher”: for a better reading, impaired scores should be highlighted somehow, and also cognitive domains should be more clearly separated. Moreover it lacks an explanation of the acronyms used and the respective references should be provided for the tests used.

            * This table has now been amended to be easier to read, with the inclusion of clearer headings, and bolded impaired scores.   

5)      Finally: I think that case 2 (KG) might not be the best case to provide. While I completely agree that MoCA or MMSE are inadequate to provide a complete and clear cognitive picture in brain tumour patients, in the case of KG MoCA fails to show cognitive impairment…but in general KG seem to show VERY mild and limited cognitive impairments, and to my opinion the scores in executive tests are unlikely to suggest a real possible impairment in capacity and decision making.

            * The authors believe KG provides a good example of the MoCA being inadequate in the assessment of neuropsychological deficits in brain tumour patients.  He scored within the “intact” range on the MoCA despite showing neuropsychological deficits in visual memory, verbal memory recall, and some executive functioning. Although KG’s impairments are not extensive, his results reflect deficits in recall of verbal and visual information, which is particularly relevant as information provided to made decisions is often presented verbally, and even providing the information in writing may not increase KG’s retention of that information.  Additionally, KG presents with disinhibition within a semantic context, suggesting he may make decisions impulsively (see paragraph from line 482 in revised manuscript for a detailed discussion of this). As a result of reviewer 1’s comments, the impact of KG’s impairments on decision making has been updated to make this clearer. 

Reviewer 2 Report

Overall, it was delightfully comprehensive, very thorough, and well-written.

Changes in cognition and decision making capacity following brain tumour resection: Illustrated with two cases

Many thanks for the opportunity to provide peer review comments for this interesting article. Overall, it was delightfully comprehensive, very thorough, and well-written. I have listed some suggestions and comments below that might help further improve this article.

Major comments:

My major comments all have to do with recommendations on what comes next, what do we do now. Even if there is very little published literature on this particular topic, it is an issue the treatment teams are well aware of and deal with on a daily basis. What are your thoughts on these points:

1.      Although it may be good to routinely assess for decision making ability and capacity in brain tumour patients, it is highly unlikely to be achievable in clinical practice due to very limited resources. However if we’d only assess those people in which decision making capacity is questioned (which should be the case now in clinical practice), we are likely to miss issues in a significant proportion of patients. What would be your suggestion or solution here?

2.      On a related note, if we were to find decision making issues in more patients, what is the right course of action? How do we support those patients better in decision making processes? Do we have the resources and staff to provide better support? If we do not, would that perhaps lead to less patients ending up in clinical trials because of their limited decision making capacity, even if participating in those trials may improve their outcomes significantly? Do we rob them of opportunities (or hope, in the very least) in that case, or is it simply unethical to recruit those patients into studies. Please comment.

3.      An important point that I read very little about in this article, is the role of family caregivers. Family members play a very important role in the clinical decision making process, they can support patients throughout this process and are often the most suited to point out when patients are about to make a decision that is uncharacteristic for them. Which could then lead to formal assessment of decision making capacity.

Please comment and/or revise the manuscript based on these major comments. Below I have listed some other suggestions for your consideration.

Other comments:

Abstract:

·        It is mentioned that recent evidence suggests that comprehensive neuropsychological assessment has greater sensitivity than screening tools. Of course it would depend on what is considered ‘recent’, but I would say this has been well-known for quite some time. Perhaps it’d be good to remove the word ‘recent’ here.

Introduction:

·        The first sentence states that cognitive changes often occur as a consequence of brain tumour resection. Although this is of course true, the cognitive changes may also occur because of the brain tumour itself, or because of other treatments patients undergo around the same time (radiotherapy, chemotherapy, corticosteroids, etc). This is covered later of course but it might be good to revise the first sentence.

·        Line 31 (first paragraph introduction); ‘including those with a brain tumour patients’ should probably read ‘including those with a brain tumour’.

·        Further on in the review I noticed brain mets were also discussed. It may be good when clarifying that you are focusing on adults, that you are discussing all primary and secondary malignant (and benign?) brain tumours in adults. This is a strong point of the article so worth emphasising.

Tumour and treatment factors affecting capacity:

·        It is stated that 1/3 of grade IV patients have decision making incapacity (reference 15). Perhaps it would be good to explain a bit more about the population studied; where these patients that had just received first treatment, were about to undergo (more) treatment, had disease progression, etc?

·        When cognitive effects of radiotherapy are discussed the focus is on whole brain radiotherapy. As the authors already state this can have big consequences for cognitive functioning. However in those with primary brain tumours it is almost never used – more targeted methods are much more common. Perhaps it would be good to make this distinction.

Medical decision making

·        One important point I am not seeing here is that of family caregivers, as also mentioned above. They are, in most cases, very much a part of the medical decision making process.

·        If we were to assess capacity routinely, what do we do with those patients who lack capacity? Often, the time between first presentation and treatment is short and there is very limited time to perform these assessments in order to provide the best standard of care. Is it acceptable, ethically, to delay this to assess decision making capacity even if there are no apparent signs that the patient lacks capacity?

Return to occupational functioning

·        While I acknowledge it is an important point the authors are making, the second part of this section is dedicated to survivors of childhood brain tumours and returning to school etc. It was stated before that this review would focus only on adults with brain tumours so I would think this part should be rewritten? Please comment.

Research participation

·        It is stated that ‘patients are crucial for medical research’. Of course this is true but the medical research that is being conducted is also very much crucial for patients – even if the treatments that are being tested may or may not prove to be beneficial, there are studies suggesting that simply being enrolled in medical research trials is associated with better patient care and outcomes.

·        Here, too, the paediatric population is discussed.

Executive functions

·        Minor point, I would end with ‘adequate’ instead of ‘intact executive functions appear essential…’. It is almost impossible to determine if EF is intact without a premorbid assessment as you will know. I realise this is just semantics, so it’s just a suggestion.

Box 2:

·        This box with clinical guidelines is not referred to in the text. Are these guidelines based on the authors’ expert opinion? It does contain new information (e.g., regarding decision making in driving) which can lead to questions for readers. Driving in particular will present challenges in this context as the rules for driving after brain tumour can be quite different between countries and is sometimes not related to medical experts’ or (neuro-)psychologists’ assessments at all.

Author Response

Reviewer #2:

This reviewer thought our paper was comprehensive, thorough, and well written.  They made three suggestions to further improve our paper:

1.      Although it may be good to routinely assess for decision making ability and capacity in brain tumour patients, it is highly unlikely to be achievable in clinical practice due to very limited resources. However if we’d only assess those people in which decision making capacity is questioned (which should be the case now in clinical practice), we are likely to miss issues in a significant proportion of patients. What would be your suggestion or solution here?

            * The paragraph (commencing at line 60 in the review manuscript) has now been amended to clarify that it is not the recommendation of the authors that capacity assessments be conducted for every patient with a brain tumour, but capacity assessments should be considered when a brain tumour patient shows risks associated with decision making compromise.  Having a brain tumour is one such risk, as is the patient expressing a decision that is contrary to what they may have been expected to choose.  Additionally, the authors have included the use of brief and tailored pre- and post-surgery neuropsychological assessment as a means to help identify cognitive risk factors for impaired decision making, as well as the role of family members in helping to identify risks.

2.      On a related note, if we were to find decision making issues in more patients, what is the right course of action? How do we support those patients better in decision making processes? Do we have the resources and staff to provide better support? If we do not, would that perhaps lead to less patients ending up in clinical trials because of their limited decision making capacity, even if participating in those trials may improve their outcomes significantly? Do we rob them of opportunities (or hope, in the very least) in that case, or is it simply unethical to recruit those patients into studies. Please comment.

            * From line 277 of the revised manuscript, we disuses how neuropsychological assessment can help to identify barriers to capacity and set these as targets for neurorehabilitation to improve their capacity to make informed decisions.  The paper has now also been updated in the conclusions (from line 823) to reiterate this point, and to note that provision of strategies to compensate for the deficit can improve patient autonomy in decision making. We also note the important role of family members in facilitating decision making (paragraph from line 211 and in the conclusion).

3.      An important point that I read very little about in this article, is the role of family caregivers. Family members play a very important role in the clinical decision making process, they can support patients throughout this process and are often the most suited to point out when patients are about to make a decision that is uncharacteristic for them. Which could then lead to formal assessment of decision making capacity.

            * Thank you for highlighting this important aspect. As mentioned above, a paragraph has been added from line 211 to discuss the important role that the family caregivers play in supporting a patient to make decisions. 

Other comments:

Abstract:

·        It is mentioned that recent evidence suggests that comprehensive neuropsychological assessment has greater sensitivity than screening tools. Of course it would depend on what is considered ‘recent’, but I would say this has been well-known for quite some time. Perhaps it’d be good to remove the word ‘recent’ here.

            * The word “recent” was removed following this suggestion.

Introduction:

·        The first sentence states that cognitive changes often occur as a consequence of brain tumour resection. Although this is of course true, the cognitive changes may also occur because of the brain tumour itself, or because of other treatments patients undergo around the same time (radiotherapy, chemotherapy, corticosteroids, etc). This is covered later of course but it might be good to revise the first sentence.

            * The first sentence of the introduction was revised to clarify changes in cognition are the result of the tumour itself and its treatment. 

·        Line 31 (first paragraph introduction); ‘including those with a brain tumour patients’ should probably read ‘including those with a brain tumour’.

            * This error has been corrected. 

·        Further on in the review I noticed brain mets were also discussed. It may be good when clarifying that you are focusing on adults, that you are discussing all primary and secondary malignant (and benign?) brain tumours in adults. This is a strong point of the article so worth emphasising.

            * It has now been updated in the abstract and the introduction (line 59) to clarify that the review focuses on all brain tumour types. 

Tumour and treatment factors affecting capacity:

·        It is stated that 1/3 of grade IV patients have decision making incapacity (reference 15). Perhaps it would be good to explain a bit more about the population studied; where these patients that had just received first treatment, were about to undergo (more) treatment, had disease progression, etc?

            * Unfortunately, the referenced paper does not provide specifics of patient characteristics, so it cannot be included in the current paper.

·        When cognitive effects of radiotherapy are discussed the focus is on whole brain radiotherapy. As the authors already state this can have big consequences for cognitive functioning. However in those with primary brain tumours it is almost never used – more targeted methods are much more common. Perhaps it would be good to make this distinction.

            * We agree that the common treatments for primary and secondary brain tumours be differentiated.  The paper has been updated (from line 120) to note focused radiation therapy is used with primary brain tumours, which reduces the risk of more global cognitive deficits. 

Medical decision making

·        One important point I am not seeing here is that of family caregivers, as also mentioned above. They are, in most cases, very much a part of the medical decision making process.

            * We agree that the family plays a significant role in supporting patients to make decisions. A paragraph to this effect has been added (from line 213). 

·        If we were to assess capacity routinely, what do we do with those patients who lack capacity? Often, the time between first presentation and treatment is short and there is very limited time to perform these assessments in order to provide the best standard of care. Is it acceptable, ethically, to delay this to assess decision making capacity even if there are no apparent signs that the patient lacks capacity?

            * As mentioned above, in the paragraph from line 273 of the revised manuscript, we disuses how neuropsychological assessment can help to identify barriers to capacity and set these as targets for neurorehabilitation to improve their capacity to make informed decisions.  The paper has now also been updated in the conclusions (from line 823) to reiterate this point, and to note that provision of strategies to compensate for the deficit can improve patient autonomy in decision making. We also note the important role of family members in facilitating decision making (paragraph from line 211 and in the conclusion). From the perspective of psychologists, informed consent is an ethical imperative. We note at a previous point that conducting neuropsychological assessment prior to surgery can help to identify cognitive risk factors for impaired decision making, which could trigger an assessment of decision making capacity.  If there are no apparent signs of decision making incapacity (including neuropsychological performance), then a capacity assessment would not be necessary. 

Return to occupational functioning

·        While I acknowledge it is an important point the authors are making, the second part of this section is dedicated to survivors of childhood brain tumours and returning to school etc. It was stated before that this review would focus only on adults with brain tumours so I would think this part should be rewritten? Please comment.

            * Following reviewer 2’s comments, the section relating to children has been removed to avoid confusing the audience as it had been mentioned the paper will focus on adult patients. 

Research participation

·        It is stated that ‘patients are crucial for medical research’. Of course this is true but the medical research that is being conducted is also very much crucial for patients – even if the treatments that are being tested may or may not prove to be beneficial, there are studies suggesting that simply being enrolled in medical research trials is associated with better patient care and outcomes.

            * We agree that medical research is crucial to improving survival rates in brain tumour patients.  The paragraph on research participation from line 195 has been updated accordingly. 

·        Here, too, the paediatric population is discussed.

            * Information relating to children has been removed to maintain the focus of the paper on the adult population. 

Executive functions

·        Minor point, I would end with ‘adequate’ instead of ‘intact executive functions appear essential…’. It is almost impossible to determine if EF is intact without a premorbid assessment as you will know. I realise this is just semantics, so it’s just a suggestion.

            * As suggested, this has now been amended. 

Box 2:

·        This box with clinical guidelines is not referred to in the text. Are these guidelines based on the authors’ expert opinion? It does contain new information (e.g., regarding decision making in driving) which can lead to questions for readers. Driving in particular will present challenges in this context as the rules for driving after brain tumour can be quite different between countries and is sometimes not related to medical experts’ or (neuro-) psychologists’ assessments at all.

            * Box 2 has been amended to remove driving, as we agree requires driving capacity requires a different assessment approach, which will differ by country.  

Round 2

Reviewer 2 Report

My comments and suggestions have been properly addressed in this revised version. I have no further comments to make.